# Learning to Optimize Tensor Programs

**Tianqi Chen**[1]    **Lianmin Zheng**[2]    **Eddie Yan**[1]    **Ziheng Jiang**[1]    **Thierry Moreau**[1]
**Luis Ceze**[1]    **Carlos Guestrin**[1]    **Arvind Krishnamurthy**[1]
[1]Paul G. Allen School of Computer Science & Engineering, University of Washington
[2]Shanghai Jiao Tong University

## Abstract

We introduce a learning-based framework to optimize tensor programs for deep learning workloads. Efficient implementations of tensor operators, such as matrix multiplication and high dimensional convolution, are key enablers of effective deep learning systems. However, current systems rely on manually optimized libraries, e.g., cuDNN, that support only a narrow range of server class GPUs. Such reliance limits the applicability of high-level graph optimizations and incurs significant engineering costs when deploying to new hardware targets. We use learning to remove this engineering burden. We learn domain-specific statistical cost models to guide the search of tensor operator implementations over billions of possible program variants. We further accelerate the search using effective model transfer across workloads. Experimental results show that our framework delivers performance that is competitive with state-of-the-art hand-tuned libraries for low-power CPUs, mobile GPUs, and server-class GPUs.

## 1   Introduction

Deep learning (DL) has become ubiquitous in our daily lives. DL models can now recognize images [23], understand natural language [38], play games [27], and automate system decisions (e.g., device placement [26] and indexing [21]). Tensor operators, such as matrix multiplication and high dimensional convolution, are basic building blocks of DL models. Scalable learning systems [1, 4, 8, 2] rely on manually optimized, high-performance tensor operation libraries, such as cuDNN, that are optimized for a narrow range of hardware devices. To optimize a tensor operator, programmers must choose from many implementations that are logically equivalent but differ dramatically in performance due to differences in threading, memory reuse, pipelining and other hardware factors. Supporting diverse hardware back-ends therefore requires tremendous engineering effort. Even on currently supported hardware, developing DL frameworks and models is limited by the set of optimized operators in libraries, preventing optimizations (such as operator fusion) that can produce unsupported operators.

This research explores the following question: can we use learning to alleviate this engineering burden and automatically optimize tensor operator programs for a given hardware platform? Our affirmative answer is based on statistical cost models we built that predict program run time using a given low-level program. These cost models, which guide our exploration of the space of possible programs, use transferable representations that generalize across different workloads to accelerate search. We make the following contributions:

- We formalize the new problem of learning to optimize tensor programs and summarize its key characteristics.
- We propose a machine learning framework to solve this problem.
- We further accelerate the optimization by $2\times$ to $10\times$ using transfer learning.

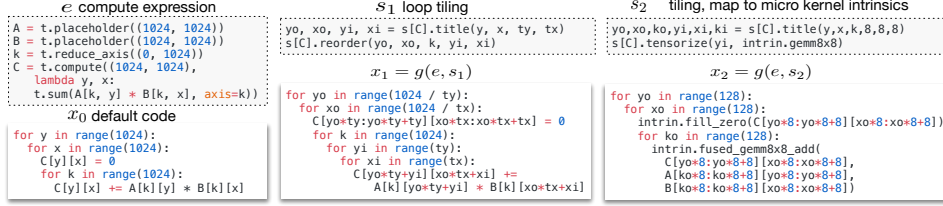

Figure 1: Sample problem. For a given tensor operator specification ($C_{ij} = \sum_k A_{ki}B_{kj}$), there are multiple possible low-level program implementations, each with different choices of loop order, tiling size, and other options. Each choice creates a logically equivalent program with different performance. Our problem is to explore the space of programs to find the fastest implementation.

We provide a detailed empirical analysis of component design choices in this framework. *Experimental results on real-world DL workloads show that our framework yields end-to-end performance improvements ranging from 1.2× to 3.8× over existing frameworks.*

## 2  Problem Formalization

We begin by walking through the motivating example in Figure 1. To enable automatic code generation, we specify tensor operators using index expressions (e.g., $C_{ij} = \sum_k A_{ki}B_{kj}$). Let $\mathcal{E}$ denote the space of index expressions. The index expression leaves many low-level implementation details, such as loop order, memory scope, and threading unspecified. As a result, we can generate multiple variants of low-level code that are logically equivalent to the expression for a given $e \in \mathcal{E}$. We use $\mathcal{S}_e$ to denote the space of possible transformations (schedules) from $e$ to low-level code. For an $s \in \mathcal{S}_e$, let $x = g(e, s)$ be the generated low-level code. Here, $g$ represents a compiler framework that generates low-level code from $e, s$. We are interested in minimizing $f(x)$, which is the real run time cost on the hardware. Importantly, we do not know an analytical expression for $f(x)$ but can query it by running experiments on the hardware. For a given tuple of $(g, e, \mathcal{S}_e, f)$, our problem can be formalized as the following objective:

$$\underset{s \in \mathcal{S}_e}{\arg\min} f(g(e, s)) \tag{1}$$

This problem formalization is similar to that of traditional hyper-parameter optimization problems [34, 33, 35, 13, 17, 25] but with several key differentiating characteristics:

**Relatively Low Experiment Cost.** Traditionally, hyper-parameter optimization problems incur a high cost to query $f$, viz., running experiments could take hours or days. However, the cost of compiling and running a tensor program is a few seconds. This property requires that model training and inference be *fast* ; otherwise, there is no benefit over profiling execution on real hardware. It also means that we can collect more training data during optimization.

**Domain-Specific Problem Structure.** Most existing hyper-parameter optimization algorithms treat the problem as a black box. As we optimize programs, we can leverage their rich structures to build effective models.

**Large Quantity of Similar Operators.** An end-to-end DL system must optimize tensor operator programs for different input sizes, shapes, and data layout configurations. These tasks are similar and can offer opportunities for transfer learning.

We describe two key prerequisites for automatic code generation that is competitive with hand-optimized code. (**1**) We need to define an exhaustive search space $\mathcal{S}_e$ that covers all hardware-aware optimizations in hand-tuned libraries. (**2**) We need to efficiently find an optimal schedule in $\mathcal{S}_e$.

There are many domain-specific languages (DSLs) for code generation [32, 36, 15, 37, 20, 30], each with with a different $\mathcal{E}$, $\mathcal{S}_e$ and $g$. Polyhedral models [5, 42, 41] are a popular choice for $\mathcal{S}_e$; they model the loop domains as integer linear constraints. An alternative approach originating from Halide [32] defines a schedule space using a set of transformation primitives. Improving $\mathcal{S}_e$ is an important research direction that is beyond the scope of this paper; we pick a rich $\mathcal{S}_e$ and focus on schedule optimization in the rest of the paper.

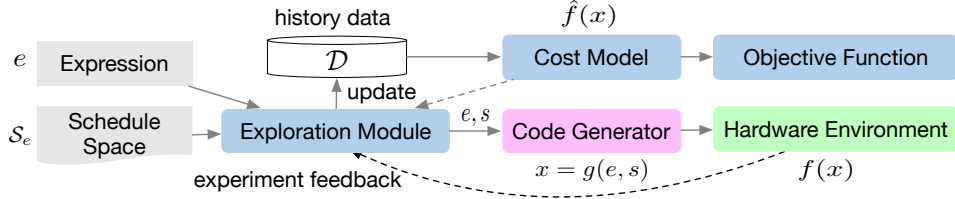

Figure 2: Framework for learning to optimize tensor programs.

We use primitives from an existing code generation framework [9] to form $\mathcal{S}_e$. Our search space includes multi-level tiling on each loop axis, loop ordering, shared memory caching for GPUs, and annotations such as unrolling and vectorization. The search space size $|\mathcal{S}_e|$ can be on the order of billions of possible implementations for a single GPU operator. As we find in section 6 , our choice of $\mathcal{S}_e$ can contain programs competitive with hand-optimized libraries.

## 3  Learning to Optimize Tensor Programs

We propose a machine learning (ML)-based framework to solve this problem. Figure 2 presents the framework and its modules. We build a statistical cost model $\hat{f}(x)$ to estimate the cost of each low-level program $x$. An exploration module proposes new schedule configurations to run on the hardware. The run time statistics are collected in a database $\mathcal{D} = \{(e_i, s_i, c_i)\}$, which can in turn be used to update $\hat{f}$. We discuss module-specific design choices in the following subsections.

### 3.1  Statistical Cost Model

The first statistical model we support is based on gradient boosted trees [11](GBTs). We extract domain-specific features from a given low-level abstract syntax tree (AST) $x$. The features include loop structure information (e.g., memory access count and data reuse ratio) and generic annotations (e.g., vectorization, unrolling, thread binding). We use XGBoost [7], which has proven to be a strong feature-based model in past problems. Our second model is a TreeGRU[39], which recursively encodes a low-level AST into an embedding vector. We map the embedding vector to a final predicted cost using a linear layer.

GBT and TreeGRU represent two distinct ML approaches to problem resolution. Both are valuable, but they offer different benefits. GBT relies on precise feature extraction and makes fast predictions using CPUs. TreeGRU, the deep learning-based approach, is extensible and requires no feature engineering, but it lags in training and predictive speed. We apply batching to the TreeGRU model and use GPU acceleration to make training and prediction fast enough to be usable in our framework.

### 3.2  Training Objective Function

We can choose from multiple objective functions to train a statistical cost model for a given collection of data $\mathcal{D} = \{(e_i, s_i, c_i)\}$. A common choice is the regression loss function $\sum_i (\hat{f}(x_i) - c_i)^2$ , which encourages the model to predict cost accurately. On the other hand, as we care only about the relative order of program run times rather than their absolute values in the selection process, we can instead use the following rank loss function [6]:

$$\sum_{i,j} \log(1 + e^{-\operatorname{sign}(c_i - c_j)(\hat{f}(x_i) - \hat{f}(x_j))}). \tag{2}$$

We can use the prediction $\hat{f}(x)$ to select the top-performing implementations.

### 3.3  Exploration Module

The exploration module controls the search loop, which is summarized in Algorithm 1. At each iteration, it must pick a batch of candidate programs based on $\hat{f}(x)$ and query $f(x)$ on real hardware. We cannot simply enumerate the entire space of $S_e$ and pick the top-b candidates due to the size of the search space. Instead, we use simulated annealing [19] with $\hat{f}(x)$ as the energy function.

---

**Algorithm 1:** Learning to Optimize Tensor Programs

---

**Input**  : Transformation space $\mathcal{S}_e$
**Output :** Selected schedule configuration $s^*$
$\mathcal{D} \leftarrow \emptyset$
**while** *n_trials < max_n_trials* **do**
  // Pick the next promising batch
  $Q \leftarrow$ run parallel simulated annealing to collect candidates in $\mathcal{S}_e$ using energy function $\hat{f}$
  $S \leftarrow$ run greedy submodular optimization to pick $(1 - \epsilon)b$-subset from $Q$ by maximizing Equation 3
  $S \leftarrow S \cup \{$ Randomly sample $\epsilon b$ candidates. $\}$
  // Run measurement on hardware environment
  **for** $s$ *in* $S$ **do**
  | $c \leftarrow f(g(e, s)); \mathcal{D} \leftarrow \mathcal{D} \cup \{(e, s, c)\}$
  **end**
  // Update cost model
  update $\hat{f}$ using $\mathcal{D}$
  n_trials $\leftarrow$ n_trials $+ b$
**end**
$s^* \leftarrow$ history best schedule configuration

---

Specifically, we use a batch of parallel Markov chains to improve the prediction throughput of the statistical cost model. We select the top-performing batch of candidates to run on real hardware. The collected performance data is used to update $\hat{f}$. We make the states of the Markov chains persistent across $\hat{f}$ updates. We also apply the $\epsilon$-greedy to select $\epsilon b$ (e.g. 0.05) candidates randomly to ensure exploration.

**Diversity-Aware Exploration.**  We consider both quality and diversity when selecting $b$ candidates for hardware evaluation. Assume that the schedule configuration $s$ can be decomposed into $m$ components $s = [s_1, s_2, \cdots s_m]$. We maximize the following objective to select candidate set $S$ from the top $\lambda b$ candidates:

$$L(S) = -\sum_{s \in S} \hat{f}(g(e, s)) + \alpha \sum_{j=1}^{m} |\cup_{s \in S} \{s_j\}| \tag{3}$$

The first term encourages us to pick candidates with low run time costs. The second term counts the number of different configuration components that are covered by $S$. $L(S)$ is a submodular function, and we can apply the greedy algorithm [29, 22] to get an approximate solution.

**Uncertainty Estimator.**  Bayesian optimization methods [34, 33, 35, 17] use acquisition functions other than the mean when an uncertainty estimate of $\hat{f}$ is available. Typical choices include expected improvement (EI) and upper confidence bound (UCB). We can use bootstrapping to get the model's uncertainty estimate and validate the effectiveness of these methods. As we will see in section 6 , considering uncertainty does not improve the search in our problem. However, the choice of acquisition function remains a worthy candidate for further exploration.

## 4   Accelerating Optimization via Transfer Learning

Thus far, we have focused only on learning to optimize a single tensor operator workload. In practice, we need to optimize for many tensor operators with different input shapes and data types. In a real world setting, the system collects historical data $\mathcal{D}'$ from previously seen workloads. We can apply transfer learning to effectively use $\mathcal{D}'$ to speed up the optimization.

The key to transfer learning is to create a **transferable representation** that is **invariant** to the source and target domains. We can then share the cost model using the common representation across domains. Different choices of representations may have different levels of invariance.

A common practice in Bayesian optimization methods is to directly use configuration $s$ as the model's input. However, the search space specification can change for different workloads or when the user specifies a new search space for the same workload. The configuration representation $s$ is not invariant to changes in the search space.

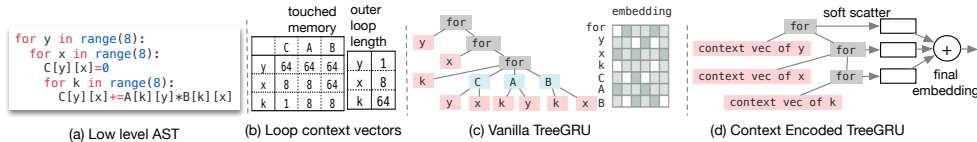

Figure 3: Possible ways to encode the low-level loop AST.

| Workload Name | C1 | C2 | C3 | C4 | C5 | C6 | C7 | C8 | C9 | C10 | C11 | C12 |
|---|---|---|---|---|---|---|---|---|---|---|---|---|
| H, W | 224,224 | 56,56 | 56,56 | 56,56 | 56,56 | 28,28 | 28,28 | 28,28 | 14,14 | 14,14 | 14,14 | 7,7 |
| IC, OC | 3,64 | 64,64 | 64,64 | 64,128 | 64,128 | 128,128 | 128,256 | 128,256 | 256,256 | 256,512 | 256,512 | 512,512 |
| K, S | 7,2 | 3,1 | 1,1 | 3,2 | 1,2 | 3,1 | 3,2 | 1,2 | 3,1 | 3,2 | 1,2 | 3,1 |

Table 1: Configurations of all conv2d operators in a single batch ResNet-18 inference. H,W denotes height and width, IC input channels, OC output channels, K kernel size, and S stride size.

On the other hand, the low-level loop AST $x$ (Figure 3a) is a shared representation of programs that is invariant to the search space. To leverage this invariance, our cost model $\hat{f}(x)$ takes the low-level loop AST $x$ as input. We also need to encode $x$ into a vector space to perform prediction. The specific encoding of $x$ can also result in different levels of invariance.

**Context Relation Features for GBT.** We define context features at each loop level to represent loop characteristics. A simple representation of context features is a vector (e.g., in Figure 3b, where each loop has a row of features). Context features are informative but, crucially, cannot generalize across different loop nest patterns; we define context relation features to overcome this issue.

To build context relation features, we treat context vectors as a bag of points and extract features that model relations between feature axes. Formally, let $Z$ be the context feature matrix such that $Z_{ki}$ corresponds to the $i$-th feature of loop $k$. We define a set of $\log_2$-spaced constant thresholds $\beta = [\beta_1, \beta_2, \cdots \beta_m]$. The relation feature between feature $i$ and $j$ is defined as: $R_t^{(ij)} = \max_{k \in \{k | Z_{kj} < \beta_t\}} Z_{ki}$. This encoding summarizes useful relations, such as loop count vs. touched memory size (related to the memory hierarchy of the access), that affect run time cost.

**Context Encoded TreeGRU.** The invariant representation also exists for the neural-based model. Figure 3c shows a way to encode the program by learning an embedding vector for each identifier and summarizing the AST using TreeGRU. This model works well for modeling a single workload. However, the set of loop variables can change across different domains, and we do not have embedding for the new loop variables. We instead encode each loop variable using the context vector extracted for GBT to summarize the AST (Figure 3d). We scatter each loop level, embedding $h$ into $m$ vectors using the rule $out_i = \text{softmax}(W^T h)_i h$. Conceptually, the softmax classifies the loop level into a memory slot in $out$. Then, we sum the scattered vectors of all loop levels to get the final embedding.

Once we have a transferable representation, we can use a simple transfer learning method by combining a global model and an in-domain local model, as follows:

$$\hat{f}(x) = \hat{f}^{(global)}(x) + \hat{f}^{(local)}(x). \tag{4}$$

The global model $\hat{f}^{(global)}(x)$ is trained on $\mathcal{D}'$ using the invariant representation; it helps to make effective initial predictions before we have sufficient data to fit $\hat{f}^{(local)}(x)$.

## 5 Prior Work

Black box optimization (auto-tuning) is used in high-performance computing libraries such as ATLAS [43] and FFTW [12]. Alternatively, a hardware-dependent cost model can be built to guide the search [28, 5]. Polyhedral methods [5, 42] use integer linear programming to optimize cost. Tensor Comprehensions [41] combine both approaches, using black-box optimization to choose parameters of thread blocks and polyhedral optimization to generate internal loops. Black-box approaches can require many experiment trials to explore a huge $\mathcal{S}_e$. On the other hand, predefined cost models may not be sufficiently accurate to capture the complexity of modern hardware and must be manually redefined for each new hardware target.

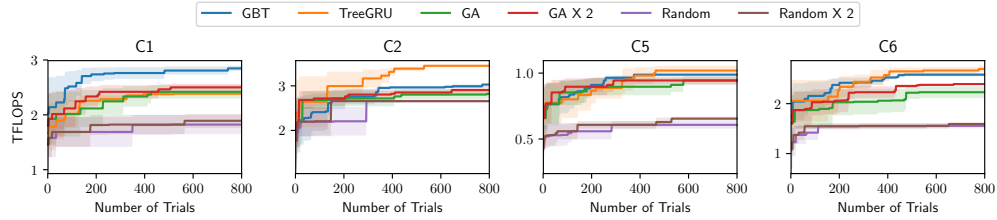

Figure 4: Statistical cost model vs. genetic algorithm (GA) and random search (Random) evaluated on NVIDIA TITAN X. 'Number of trials' corresponds to number of evaluations on the real hardware. We also conducted two hardware evaluations per trial in Random ×2 and GA ×2. Both the GBT- and TreeGRU-based models converged faster and achieved better results than the black-box baselines.

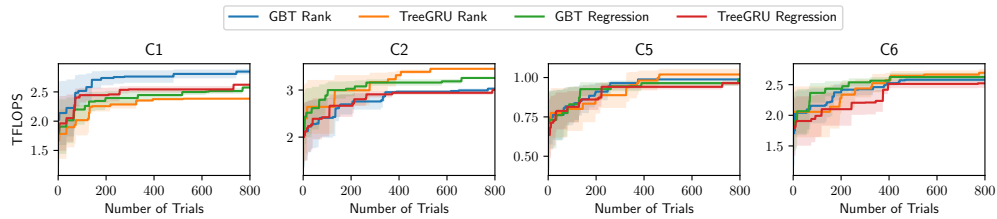

Figure 5: Rank vs. Regression objective function evaluated on NVIDIA TITAN X. The rank-based objective either outperformed or performed the same as the regression-based objective in presented results.

Previously, statistical cost models have been applied to optimize SAT solvers [17, 18]. We apply this idea to our problem and build a domain-specific cost model that enables effective transfer among workloads. A recent trend is to use deep neural networks to perform program analysis [3, 10]. Our new problem setting and experiment environment can serve as a testbed for unexplored research opportunities in related directions.

# 6 Experiments

## 6.1 Component Evaluations

We first evaluated each design choice in the framework. Component evaluations were based on convolution workloads in ResNet-18 [14] for ImageNet classification (Table 1). Due to space limitations, we show component evaluation results only on representative workloads; the complete set of results is reported in the supplementary material. All methods compared in this subsection were initialized with no historical data. Section 6.2 evaluates the transfer learning setting.

**Importance of Statistical Cost Model.** Figure 4 compares the performance of the statistical cost model to black-box methods. Both the GBT and TreeGRU models outperformed the black-box methods and found operators that were 2× faster than those found with random searches. This result is particularly interesting compared to prior results in hyper-parameter tuning [25], where model-based approaches were shown to work only as well as random searching. Our statistical models benefit from domain-specific modeling and help the framework find better configurations.

**Choice of Objective Function.** We compared the two objective functions in Figure 5 on both types of models. In most cases, we found that using a rank-based objective was slightly better than using a regression-based one: the rank-based objective may have sidestepped the potentially challenging task of modeling absolute cost values. We chose rank as our default objective.

**Impact of Diversity-Aware Exploration.** We evaluated the impact of the diversity-aware exploration objective in Figure 6. Most of the workloads we evaluated showed no positive or negative impact for diversity-based selection. However, diversity-aware exploration improved C6, which shows some potential usefulness to the approach. We adopted the diversity-aware strategy since it can be helpful, has no meaningful negative impact, and negligibly affects run time.

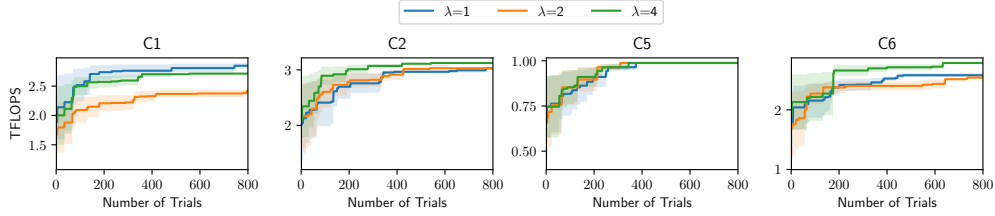

Figure 6: Impact of diversity-aware selection with different choices of $\lambda$ evaluated on NVIDIA TITAN X. Diversity-aware selection had no positive or negative impact on most of the evaluated workloads.

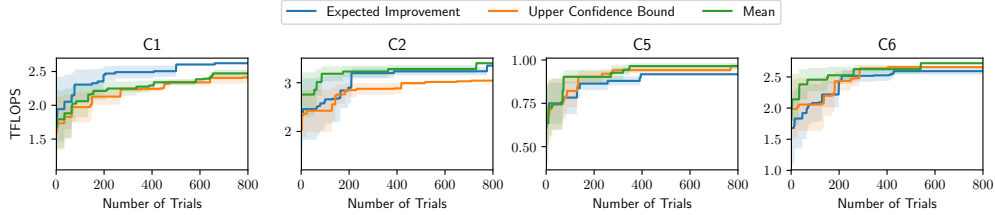

Figure 7: Impact of uncertainty-aware acquisition functions evaluated on NVIDIA TITAN X. Uncertainty-aware acquisition functions yielded no improvements in our evaluations.

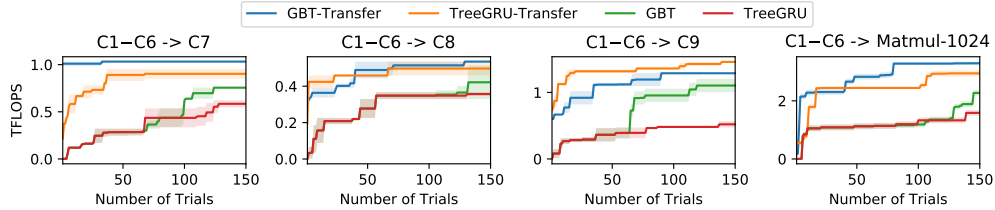

Figure 8: Impact of transfer learning. Transfer-based models quickly found better solutions.

**Impact of Uncertainty Estimator.**   We evaluated the usefulness of uncertainty-aware acquisition functions in Figure 7. The uncertainty measurement was achieved by training five models using bootstrapping. We used the regression objective in this setting—similar to its use in most Bayesian optimization methods. Results show that uncertainty estimation was not as important in our problem, possibly because our models were trained with more training samples than traditional hyper-parameter optimization problems.

### 6.2   Transfer Learning Evaluations

The evaluations presented so far used no historical data. This subsection evaluates the improvements obtainable with transfer learning.

**Improvements by Transfer.**   We first evaluated general improvements made possible by transfer learning. We randomly picked samples from $\mathcal{D}'$ collected from C1,C2,C3,C4,C5,C6 and used them to form the source domain (30000 samples in the TITAN X experiment and 20000 samples in the ARM GPU and ARM A53 experiments). We then compared the performance of transfer-enabled methods to learning from scratch for target workloads C7,C8,C9. Results are shown in Figure 8. Overall, using transfer learning yielded a $2\times$ to $10\times$ speedup. This approach is especially important for real DL compilation systems, which continuously optimize incoming workloads.

**Invariant Representation and Domain Distance.**   As discussed in Section 4, different representations have different levels of invariance. We used three scenarios to study the relationship between domain distance and the invariance of feature representations: (1) running optimizations on only one target domain; (2) C1–C6→7: C1–C6 as source domain and C7 as target domain (transfer within same operator type); (3) C1–C6→Matmul-1024: C1–C6 as source domain and matrix multiplication as

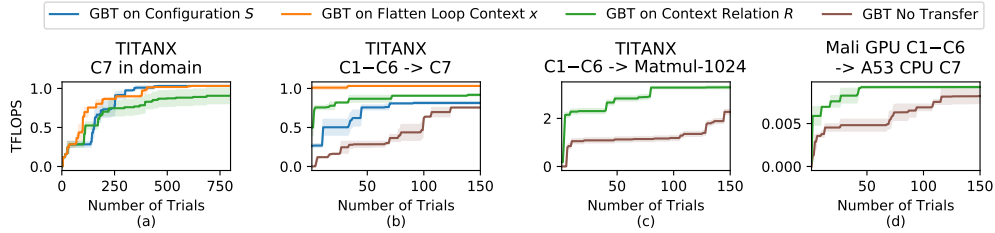

Figure 9: Comparison of different representations in different transfer domain settings. The configuration-based model can be viewed as a typical Bayesian optimization approach (batched version of SMAC [17]). We found that models using configuration space features worked well within a domain but were less useful across domains. The flattened AST features worked well when transferring across convolution workloads but were not useful across operator types. Context relation representation allowed effective transfer across operator types.

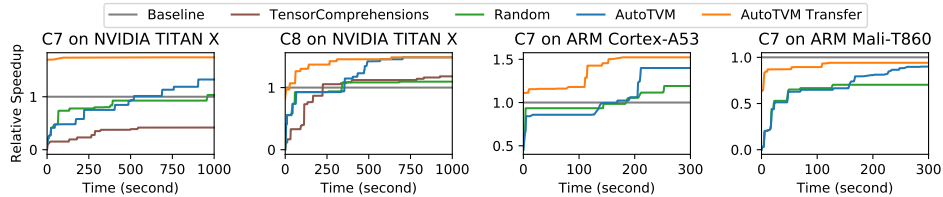

(a) Optimization curves in wall clock time. (We set cuDNN v7, Tensorflow Lite and ARM ComputeLibrary v18.03 as the baselines for TITAN X, ARM A53 and ARM Mali-T860, respectively.)

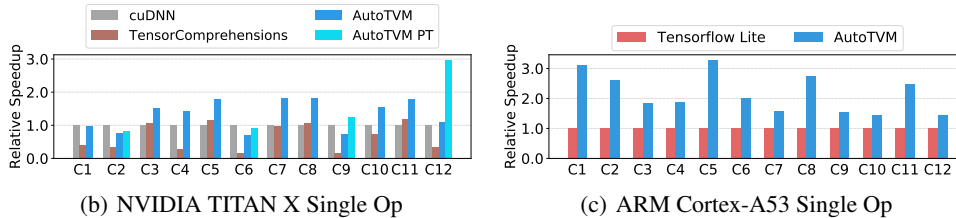

(b) NVIDIA TITAN X Single Op

(c) ARM Cortex-A53 Single Op

Figure 10: Single operator performance on the TITAN X and ARM CPU. (Additional ARM GPU (Mali) results are provided in the supplementary material.) We also included a weight pre-transformed Winograd kernel [24] for $3 \times 3$ conv2d (AutoTVM PT). AutoTVM generated programs that were competitive with hardware-specific libraries.

target domain (transfer across operator types). Results ( Figure 9) show the need for more invariance when domains are farther apart. Using our transferable feature representation, our model generalized across different input shapes and operator types. We also ran a preliminary study on transfer from an ARM Mali GPU to an ARM Cortex-A53 ( Figure 9d), showing that the proposed representation enabled transfer across devices. Developing an invariant feature representation poses a difficult problem worthy of additional research.

## 6.3 End-to-End Evaluation

Thus far, our evaluation has focused on specific design choices in our framework. We now segue to the natural follow-up question: can learning to optimize tensor programs improve real-world deep learning systems on diverse hardware targets? We call our framework AutoTVM. We compared our approach to existing DL frameworks backed by highly engineered hardware-specific libraries on diverse hardware back-ends: a server class GPU, an embedded CPU, and a mobile GPU. Note that AutoTVM performs optimization and code generation *with no external operator library*.

We first evaluated single-operator optimization against baselines that used hardware-specific libraries. The baselines were: cuDNN v7 for the NVIDIA GPU, TFLite(commit: 7558b085) for the Cortex-A53, and the ARM Compute Library (v18.03) for the ARM Mali GPU. We also in-

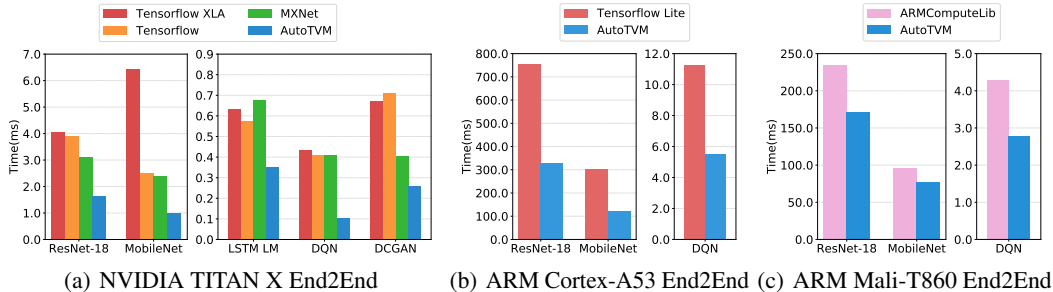

(a) NVIDIA TITAN X End2End  (b) ARM Cortex-A53 End2End (c) ARM Mali-T860 End2End

Figure 11: End-to-end performance across back-ends. [2]AutoTVM outperforms the baseline methods.

cluded TensorComprehensions (commit: ef644ba) [41] as an additional baseline for the TITAN X [1] TensorComprehensions used 2 random seeds $\times 25$ generations $\times 200$ population for each operator, and padding was removed (TC does not yet support padding). The results are shown in Figure 10. AutoTVM generated high-performance tensor programs across different hardware back-ends.

Further, we embedded our framework into an existing DL graph compiler stack and performed end-to-end workload evaluation. We evaluated real world end-to-end DL inference workloads, including ResNet [14], MobileNet [16], LSTM Language Model [44], Deep Q Network (DQN) [27], and Deep Convolutional Generative Adversarial Networks (DCGAN) [31]. Our baselines were: MXNet (v1.1), Tensorflow (v1.7) for the GPU, TFLite(commit: 7558b085) for the Cortex A53, and ARM Compute Library (v18.03) for the ARM Mali GPU. Results are summarized in Figure 11. AutoTVM improved end-to-end performance by $1.2\times$ to $3.8\times$. These improvements were due to both tensor program optimization and operator fusion optimizations; the latter would otherwise be impossible if we used libraries with a limited set of operators.

## 7 Discussion and Conclusion

We presented AutoTVM: a machine learning-based framework that automatically optimizes the implementation of tensor operators in deep learning systems. Our statistical cost model allows effective model sharing between workloads and speeds up the optimization process via model transfer. The positive experimental results of this new approach show promise for DL deployment. Beyond our solution framework, the specific characteristics of this new problem make it an ideal testbed for innovations in related areas, such as neural program modeling, Bayesian optimization, transfer learning, and reinforcement learning. On the systems side, learning to optimize tensor programs can enable more fused operators, data layouts, and data types across diverse hardware back-ends—crucial to improving DL systems. Our framework can be found at https://tvm.ai.

### Acknowledgement

We would like to thank members of Sampa, SAMPL and Systems groups at the Allen School for their feedback on the work and manuscript. This work was supported in part by a Google PhD Fellowship for Tianqi Chen, ONR award #N00014-16-1-2795, NSF under grants CCF-1518703, CNS-1614717, and CCF-1723352, and gifts from Intel (under the CAPA program), Oracle, Huawei and anonymous sources.

## Footnotes

[1]According to personal communication [40], TC is not yet intended for use in compute-bound problems. However, it still provides a good reference baseline for inclusion in the comparison.

[2]DCGAN and LSTM were not reported on A53 and Mali because they are not yet supported by baseline systems.

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
