[Supplementary Material]

# Learning to Optimize Tensor Programs: Supplementary Materials

## A Supplementary Materials

### A.1 Additional Experimental Results

Figure 1: Single Operator Performance on Mali T860MP4

Figure 2: Effectiveness of cost model on all conv2d operators in ResNet-18.

Figure 3: Impact of objective function of cost model on all conv2d operators in ResNet-18.

Figure 4: Impact of diversity aware exploration on all conv2d operators in ResNet-18.

Figure 5: Impact of uncertainty aware acquisition function on all conv2d operators in ResNet-18.

## A.2 Summary of Loop Features

### A.2.1 Loop Context

We extract loop context for every loop variable. The loop context contains loop attributes and the access patterns for all touched inner buffers.

| Feature Name | | Description |
|---|---|---|
| length | | The length of this loop |
| annotation | | One-hot annotation of this loop (can be vectorize, unrolled, paralleled, ...) |
| top-down | | The product of the lengths of outer loops |
| bottom-up | | The product of the lengths of inner loops |
| access pattern (for every buffer) | touch count | The number of touched elements |
| | reuse ratio | Reuse ratio of this buffer (= bottom-up / touch count) |
| | stride | Coefficent of this loop varialbe in the index expression |

Table 1: Listing of loop context feature

### A.2.2 Relation Feature

First we pick the longest chain from the AST. Then we extract loop context features for the loop variables in this chain. We compute two pairs of relation : touch count vs reuse ratio and touch count vs top-down.

### A.3 Experiment Configuration

| Hyperparameter | Value | Description |
| --- | --- | --- |
| $b_{GBT}$ | 64 | batch size of planning in GBT |
| $b_{TreeGRU}$ | 64 | batch size of planning in TreeGRU |
| $emb\_dim$ | 128 | dimension of loop variable embedding in TreeGRU |
| $hidden\_size$ | 128 | hidden size of GRU cell in TreeGRU |
| $n_{sa}$ | 128 | number of Markov chains in parallel simulated annealing |
| $step_{sa}$ | 500 | maximum steps of one simulated annealing run |