[Reviews · NeurIPS 2018]

Reviewer 1



The paper presented a pipeline that is able to effectively search in a program-configuration space for more performant tensor programs, under various operator types and hardware architectures. Although methodology wise this paper doesn't introduce novel techniques, I still find the overall execution (i.e. piecing together the many different components with sensible choices of techniques and demonstrating convincing empirical improvements through carefully designed experiments) impressive and believe it will have a tangible impact. Detailed comments: * When you say "Each choice creates a logically equivalent program ..." under Figure 1, is it because, **by design**, none of the configurations would change the logic of the program (in which case I think this worth a more explicit emphasis)? * The actual search space is missing a more detailed and formal breakdown (except from the verbal descriptions on L.75 - 76 on P.2). * I'd like to see a more direct evaluation of the performance of the learnt statistical cost model itself (local or global), i.e. in terms of metrics like training/test error, ranking loss, etc. * The way the estimated running costs are used in Eq.(3) (the summation and the balancing against "coverage" with alpha) seems to suggest the actual scales of the cost values do matter in this case, while the ranking loss in Eq.(2) is shift-invariant, which may introduce a slight discrepancy in the modeling setup (and hence partially contribute to why this "diversity-aware" exploration didn't play out as effectively as expected)? Also, how is alpha decided? * When you say "the search space specification can change for different a workload" in L.136 on P.4, it's not immediately clear within the context indicated by Table 1 why it should be the case. If you are implying a different type of workload (e.g. "Matmul"?) here please make it clearer. * In Eq.(4), is the global model being kept fixed during transfer learning or is it being updated alongside the local model? * 2 curves seem missing in Fig.9 (c) and (d), and as a result, the claim that "The flattened AST features ... operator types" seems ungrounded. * In Fig.10 (a), none of the results have improved over the baseline for C7 on ARM Mali-T860. Any explanation why? * "operator fusion" is mentioned twice in the paper without much explanation nor reference. Also it would be nice to have a breakdown of the speed improvement with/without "operator fusion" just to have a more detailed picture.

Reviewer 2



This paper proposes to optimize tensor programs, specifically, perform automatic implementation generation for a certain set of Tensor programs (declared by users) in order to improve the execution efficiency over the (kernel) implementation provided by existing tensor computation libraries. The problem is interesting and the framework proposed in this paper seems to be technically sound. The evaluation of this paper contains three parts of results: (1) empirical studies for the specific design choice of each component of the framework (2) verifying the improvement of transfer; (3) improvement on end-to-end DL workloads. The experiments reveal that the proposed framework can result in considerable improvement on efficiency. My major concern about this paper is how useful it will be compared to using a static tensor library. Per my understanding, given a specific tensor program and a hardware configuration, regardless of whether transferring is enabled or not, the proposed framework has to perform substantial training for each parameterized modules in order for it to work, which I believe will yield substantial overhead compared to the improvement over static libraries that the framework can bring. How much exactly is the cost? Also, once the hardware changed or the tensor program changed, the previously built model on a specific program or a specific hardware environment is not adaptable, meaning the parameters probably need to be re-trained. While static tensor computation libs (e.g. CUDA, MKL) usually are optimized for a certain (board) group of hardware and software programs (i.e., therefore, one can always anticipate a certain amount of benefits by using them, as long as within its scope, such as CUDA for most GPUs). An argument made by the authors is that the “Learn to optimize Tensor Programs” frameworks can enable the automatic generation of some math operations that are not covered by de facto tensor computing libs. But how many of these operations exist and how often are they used? My experience with deep learning is that nowadays the development of DL models and the deployment of production-level DL code/workloads have been standardized a lot and those operators that are less optimized are rarely explored. While “Learning to optimize tensor program” offers some automation on generating better kernel implementation for some long-tail operators, it is questionable how useful it is in practice. - Other questions The current framework is built by collecting data through experimental trials, and learning a cost model based on it and then minimizing certain objectives given the cost model. I am wondering whether it is possible to build the framework based on reinforcement learning? The current framework seems to be built to be friendly with inference, Will it be useful for training, and how useful it will be (considering the problem I raised in my previous comments) - Writing: My second major concern about this paper is about this presentation. This current writing assumes a lot of out-of-domain contextual knowledge, which tends to prevent most NIPS audiences from clearly understanding technical details. Below are a few places. There are more places that need to be clarified. *L26: What do you mean by “preventing kernel fusion that can produce unsupported operators” *L42: Why we need to enable automatic code generation? I feel some context is missing here. Per my understanding the goal of learning to optimize a tensor program is to eventually be able to automatically generate code; you probably want to point out it for the context. *L46-47, what is S_e and what does s \in S_e specifically map to? It would be good to define it first and provide more details and context. *L87: what is AST? *L102: Could you clarify in L102 what is the “selection process” and why “in the selection process, we only care about the relative order of the run times of programs rather than their absolute values”? *In section 5 experiment, could you explain in detail for each figure what metrics in Y axis are and how they are generated (specifical experimental configurations)? For example, in Fig 5 - 7, how are the TFLOPS values generated? Are they the number of TFLOPS of a learned implementation compared to random implementations? *It would be good to provide more detailed experimental configurations in sec 6.3 about the NNs and the details of the figures. In Figure 11, is the time in ms (in Y axis) for a single inference pass? - Post rebuttal After reading the authors' response and the other two reviewers' comments, I am willing to adjust my score to be 1 point higher (from 5 to 6). The authors partially answered some of my questions about the adaptability of this work; While I am not fully convinced on this will be a competitive alternative to libraries such as CUDA/cuDNN (due to hardware-specific optimizations and running cost of the data-driven approaches), I think this work poses an interesting problem and a sensible solution to it. For concerns (Reviwer 1 and myself) on presentation, I think it could be improved in the camera-ready version (and I would urge the authors to do so).

Reviewer 3



The paper re-introduces a formalization and framework to apply machine learning to optimize Tensor operations, and evaluates it for deep learning workloads. The formalization is familiar if one reads HPC papers on automatic code generation using heuristics. However, reiterating this formalization and fitting it to deep learning workloads such as Tensor operations is appreciated. The formalization is based on: - expression: describes the computation - schedule: maps the computation onto hardware - cost: runtime of the {expression, schedule} on the hardware The authors' goal is to apply statistical learning methods to the problem -- in particular, Gradient Boosted Trees and TreeGRU neural networks. Because Gradient Boosted Trees need input featurization, the authors first propose extracting features on the low-level loop AST (a standard low-level AST) as the features to the GBT. Typical features such as "loop length" and "touched memory" are used. Because the authors want to learn low-parametric statistical models, they need features that generalize across loop nesting patterns, so they define these features in a relative manner (eg. in "for i for j for k for l", features have a chance of similarity across ij, jk, kl). The relative weighting is log-spaced, capturing the loop nesting spacing correctly. Using these features, the authors define and train a GBT and a TreeGRU. Additionally, the authors see how their models perform in a transfer-learning setting. Finetuning for out-of-domain, or domain-distant samples and seeing how the performance + learning time fare. The results are very well calibrated, with strong and robust baselines. For example, they make sure their baseline is the latest CuDNN version, TensorFlow, Tensor Comprehensions and ARMComputeLib. The results look very strong, often outperforming the baselines, and by a significant amount. This paper convincingly introduces modern neural network techniques (not just small MLPs) and GBTs to optimize code generation, in particular for deep learning workloads. It also introduces featurizers for the loop ASTs that are more suited for statistical models. In convincingly showing that there is feasibility in approaching these problems from a statistical parametric learning perspective (apart from the traditional Genetic optimization, polyhederal compilation, integer search), this paper will become a catalyst for the deep learning community to attack this problem space effectively. Hence, I think this paper is one of the top 50% (or even one of the top 10%) papers I've reviewed for NIPS this year.